# Root Foraging Precision of *Pinus pumila* (Pall.) Regel Subjected to Contrasting Light Spectra

**DOI:** 10.3390/plants10071482

**Published:** 2021-07-19

**Authors:** Chunxia He, Jun Gao, Yan Zhao, Jing Liu

**Affiliations:** 1Key Laboratory of Tree Breeding and Cultivation of National Forestry and Grassland Administration, Research Institute of Forestry, Chinese Academy of Forestry, Beijing 100091, China; hechunxia08@126.com; 2Co-Innovation Center for Sustainable Forestry in Southern China, Nanjing Forestry University, Nanjing 210037, China; 3College of Horticulture and Plant Production, Henan University of Science and Technology, Luoyang 471002, China; ccliujing@163.com

**Keywords:** *Pinus pumila*, alpine ecosystem, root foraging, fine root proliferation

## Abstract

Root foraging behavior in heterogeneous patterns of soil nutrients is not well understood for undergrowth in alpine forests, where light spectra may generate an interactive effect on root foraging precision. A dwarf alpine species, *Pinus pumila* (Pall.) Regel., was cultured in pots where nitrogen (N)–phosphorus (P)–potassium (K) nutritional granules (N–P_2_O_5_–K_2_O, 14–13–13) were added to both halves of an inner space at a rate of 67.5 mg N (homogeneous) or 135 mg N to a random half (heterogeneous). Potted seedlings were subjected to either a green-and-blue light spectrum with a red-to-green light ratio of 4.24 (15.3% red, 64.9% green, and 19.8% blue) or a red-light enriched spectrum (69.4% red, 30.2% green, and 0.4% blue) both at irradiations of 200.43 µmol m^−2^ s^−1^. The root foraging precision was assessed by the difference in the fine root morphology or weight between the two halves. The foraging precision was assessed by both fine root length and surface area and was promoted in seedlings subjected to the heterogeneous pattern in the red-light enriched spectrum. Seedlings subjected to the green-and-blue light spectrum showed lower shoot growth, biomass, and root morphology but had higher shoot and root N and P concentrations. The heterogenous pattern resulted in greater seedling growth and fine root morphology as well as N and P concentrations compared to the homogeneous pattern. We conclude that *P. pumila* has a strong ability to forage nutrients in heterogenous soil nutrients, which can be further promoted by a spectrum with higher red-light proportions.

## 1. Introduction

Light is one of the most vital environmental factors for plants. In forest ecosystems, light availability changes along a gradient of transmitted sunlight in canopy gaps that are formed by regional climate, natural succession, and herbivores [1,2,3]. Understory plants acclimate to different types of natural light environment, for example, sun-adapted species dwelling in full sunlight or shade-tolerant species in sunlight transmittance [4]. The understory light condition not only varies in light intensity but also in light quality [5,6]. Field studies have verified that sunlight spectra can influence foliar physiology and secondary metabolisms in shoots of understory dwellers [7,8]. Compared to the aerial parts of a plant, much less is known about the effects of understory light spectra on underground response.

A light spectrum could theoretically modify the underground organ growth of tree plants. When desired illumination conditions are met, lighting spectra could modify root growth by adjusting carbohydrate assimilation and dry mass allocation towards underground organs [7,9]. Light can also promote the assimilation, synthesis, and accumulation of pigments at special wavelengths, which further adjust the internal cycling of nutrient allocation and cause root proliferation [10,11]. Spectra with high wavelengths over 600 nm can facilitate root growth in herbal and crop plants [10,11,12]. The roots of woody plants even show more plasticity to light spectra than shoots across tree and shrub species [13,14]. Spectrums enriched in the red-light bandwidth (600–700 nm) either caused an increase [15,16,17] or imposed a null effect on root biomass [18,19,20,21]. Root morphology showed contrasting responses to red and blue-and-green light (<600 nm)-enriched spectra [14,17,22]. Current findings about the light spectra effect on roots are mainly derived on the hypothesis that the rhizosphere is a homogeneous space where nutrients are evenly distributed. However, the pattern of soil nutrient distribution in the field is usually heterogeneous, which generates high uncertainty in root development and proliferation as foraging responses [23,24]. Little is known about the effect of light spectra on the precision of root foraging in heterogeneous soil environments.

The heterogeneous distribution of soil nutrients results from uneven organic inputs, diverse microbial communities, and inorganic ion diffusion from decomposition [25,26]. Plant roots proliferate in nutrient-enriched soil patches [24,27,28,29]. The placement of a lateral fine root system in micro-scale soils usually presents a heterogeneous pattern as well. Root foraging behavior can be assessed by the parameters of “scale” and ”precision”. The foraging scale is defined as the mass and extent of a root system produced in a given unit of time [27]. It varies from centimeters to meters depending on the geographical scale of the subjects’ distribution [30]. A root foraging scale is usually employed to measure the capability of root expansion in a given underground space for coexisting plants at the community level or larger [28,29,30,31]. At the microsite scale, foraging precision is usually used to quantify the ability to place growing roots in nutrient enriched soil patches [27,29]. Therefore, foraging precision is a key variable for assessing the response of the root foraging ability of understory plants subjected to different light spectra. To our knowledge, there is little documented data about light quality effect on the precision of root foraging in heterogeneous soil environments.

*Pinus pumila* (Pall.) Regel is a type of dwarf-pine species that looks like an alpine shrublike tree. It is mainly distributed in the alpine zone of Japan, eastern Siberia, mid-west Europe, and Northeast China [32,33]. At low altitudes (800 m a.s.l.) of alpine ecosystems, *P. pumila* grows as an undergrowth dweller to tall-canopy trees [34]. It is an extremely slow-growing perennial [33]. The annual shoot elongation of *P. pumila* starts with an increase in air temperature during summertime from early June to late July [35,36,37]. This time also synchronizes with the fast growth of tall-tree crowns in forests with *P. pumila* populations [35,36]. Although *P. pumila* can live in forests at low altitudes, they cannot tolerate even slight shading [38]. As a result, their populations concentrate in large forest gaps or better illuminated slopes while individuals shaded by tree crowns appear suppressed, such as in the mountains in the Russian Far East [38]. However, *P. pumila* populations in central Kamchatka of Russia and northern parts of Japan are also regarded as a shade-tolerant species [34,39]. Understory light quality was found to vary in different forests of temperate montane ecosystems [7,8]. *Pinus pumila* needles contribute to photosynthesis as a response to irradiance [40]. Regarding the highly heterogenous pattern of soil nutrient availabilities in alpine forests [41], the root foraging behavior of *P. pumila* may partially account for individual growth performance in under-crown populations. The complexity of sunlight quality transmittance increases uncertainty in predicting the root foraging precision of *P. pumila*.

In this study, we conducted a bioassay study by raising potted *P. pumila* seedlings in a controlled environment where light spectra and heterogeneous nutrient pattern were simulated based on field conditions. The objective was to detect the foraging precision of this species in response to combined light spectra and soil heterogeneity, and the physiological mechanism of this result. According to existing findings, we hypothesized (i) that different light spectra can result in varied root proliferation in heterogeneous nutrient patterns and (ii) that red-light enriched spectrum induces higher root foraging precision compared with other types of spectra.

## 2. Materials and Methods

### 2.1. Plant Materials

Seeds of *P. pumila* were collected from mother trees in natural populations at low altitude (800–900 m a.s.l.) in the Great Khingan Mountains (51°39′–51°54′ N, 121°36′–122°07′ E). Before sowing, the seeds were kept in wet sands at about −25 °C for two years. The seeds were germinated in containers with wet sands for 3.5 months. Germinated seeds were transplanted to a growing chamber, where temperature and relative humidity (RH) were kept at 28 ± 2 °C and 80 ± 5%, respectively. The lateral roots of germinated seedlings were excised, leaving 45% of their initial length. When seedlings grew to have stems roughly 2.3 cm in length with a root-collar diameter (RCD) of about 0.8 mm, evenly sized individuals were transferred to 0.45-L plastic pots (top diameter × bottom diameter × height, 11.5 cm × 7.5 cm × 9.5 cm). During the experiment, seedlings were raised with growing media of mixed peat and perlite in volumetric proportions of 3:1 (v/v) (Mashiro-Dust^TM^, Zhiluntuowei Agric. & For. Sci. & Tech. Inc., Changchun, China). The indoor micro-environment was adjusted to ranges of temperature between 17 and 36 °C (day and night) and at an RH of 72–94%.

### 2.2. Set-Up of Heterogeneous Condition

Substrates were placed at the bottom of the pots at a depth of 4 cm. A plastic barrier (~8 cm in depth) was inserted into substrates to divide the inner space of a pot into two halves. The barrier had an echelon shape with measurements of ~11.5 cm height and ~8 cm depth, which can prevent any lateral movement of fine roots or mineral nutrients with water. The variation in homogeneous and heterogeneous patterns of nutrient availabilities was created by adding different amounts of fertilizers to each pot half. In the homogeneous pattern, both halves received nitrogen (N), phosphorus (P), and potassium (K) nutrients by broadcasting 67.5 mg N of controlled-release fertilizer (CRF) granules (N–P_2_O_5_–K_2_O, 14–13–13, micro-nutrients added; No.5 Osmocote^®^, The Scotts Co., Marysville, OH, USA). In the heterogeneous pattern, a random half received 135 mg N of CRF granules, while the other half received nothing. The amount and arrangement of nutrient inputs were adapted from Wei et al. [24] to simulate the heterogeneous pattern in natural soils. Therefore, both patterns received the same total amount of 135 mg N per pot. Fertilizers were broadcasted to the surface of substrates at a depth of 4 cm. After fertilizing, the remaining pot space was filled by growing substrates up to the barrier top-edge. A transplanted seedling was placed in the middle of the top-edge of the barrier with lateral roots evenly distributed in two halves. Growing substrates were continuously added to the pot until the whole pot space was filled up.

### 2.3. Light Spectra Treatment

Different light spectra were supplied by light-emitting diode (LED) panels (length × width × height, 1.2 m × 0.4 m × 0.06 m) (Pudao Photoelectricity, Zhiluntuowei A&F S&T Inc., Changchun, China). Panels were fixed to the ceilings of three growing chambers, each of which were 1.5 m (length) × 0.5 m (width) × 0.5 m (height). Three chambers were stacked on an iron shelf.

One hundred LEDs were embedded to the surface of a panel at a spacing of 2 cm × 2 cm. Diodes were designed to emit lights in a random bandwidth of three ranges of 400–500 nm (blue), 500–600 nm (green), and 600–700 nm (red). Electric flow of red light to each panel was controlled by a 200-W transformer, and green and blue lights were controlled by a 135-W transformer. A previous investigation in the mountains (51°39’–51°54’ N, 121°36’–122°07’ E) of Northeast China determined that the photosynthesis photon flux rate (PPFD) ratio among different lights in understory transmittance was a fixed factor that fluctuated to a small extent [7,8]. We used the general ratio of green to red lights (R/G) across forests as the objective critical value (~5.0). Meanwhile, electric flows were adjusted for the two transformers to achieve an R/G ratio of 4.24, falling in the range of natural spectra and getting closest to the critical value of 5.0 [8]. Illumination at this state had a green-and-blue light-enriched light spectrum (GB) with 30% electric flow for red light and with 100% electric flow for green and blue lights. This spectrum has a total PPFD of 200.23 µmol m^−2^ s^−1^ 25 cm beneath the panel, which contains 15.3% red light, 64.9% green light, and 19.8% blue light. The contrasting light characteristics were derived by employing a red-light enriched spectrum (R), which was created using 70% for red-light electric flow and 10% for green-and-blue-light electric flow. The PPFD was 200.43 µmol m^−2^ s^−1^ and contained 69.4% red light, 30.2% green light, and 0.4% blue light. These two types of light spectra shared a highly similar PPFD, which is the basis for preconditioning the comparison of different spectra [7,10,19,20,21]. Previously, a red-enriched spectrum was successfully used for the culture of larch seedlings [15]. The spectral characteristic curves for the two types of spectra are shown in Figure 1. No natural sunlight was allowed to touch the seedlings so that artificial illumination was the only source of lighting. The daily photoperiod was set to be 18 h from 6:00 a.m. to 24:00 p.m. [42].

### 2.4. Experiment Design and Arrangement

The experiment was arranged as a split-plot design. The main block was the two types of lighting spectra, which were nested to the two nutrient-distribution patterns within sub-blocks. Seedlings were watered by a sub-irrigation system in tanks. Two tanks (inner length × width × height, 54 cm × 34 cm × 7 cm) were placed at one floor and ten pots of seedlings were arranged in one tank. One tank of ten potted seedlings was subjected to heterogeneous nutrient patterns, and the other was subjected to homogeneous patterns. All pots of seedlings in one tank were taken as a basic sampling unit, and three floors were taken as three combined-treatment replicates. All tanks were watered twice a week to a depth of 3 cm. The pots and tanks were rearranged after every watering to eliminate the edge effect.

### 2.5. Sampling and Measurements

The seedlings were cultured for three months, matching the period of annual shoot elongation of *P. pumila* in natural populations [36,38,40]. All seedlings were sampled at the end of the experiment. When sampling, ten seedlings from one tank were randomly divided into two equal groups. All seedlings were measured for shoot height and RCD. Aerial organs were excised from the pot. Five shoots from a group were used for dry weight measurement, and the other five were used for the determination of foliar variables. The root systems were sampled for three components. Tap and coarse (≥ 1 mm in diameter) roots were excised from fine roots (<1 mm in diameter) by cutting lateral roots in substrates. Fine roots in both halves of pots were carefully sampled with an intact system. In pots with a homogeneous nutrient pattern, the fine roots from both halves were labeled as medium availability as they both received 67.5 mg N per half. In pots with a heterogeneous nutrient pattern, fine roots from the half with no fertilizer input were labeled as low availability and fine roots from the other half were labeled as high availability (135 mg N per half).

Dry mass was weighed after being desiccated in an oven at 70.0 °C for 72 h. Dried samples were excised first and ground for the determination of total N and P concentrations using the Kjeldahl method and the inductively coupled plasma optical emission spectrometer instrument (Vista-MPX, Varian Inc., Palo Alto, USA) [15]. Fresh needles were used to determine chlorophyl, carotenoid and protein concentrations, and glutamine synthetase (GS) and acid phosphatase (ACP) activities. Chlorophyl-a and -b and carotenoid were determined by bathing (Type HHS, Boxun Industry Inc., Shanghai, China) in 65 °C dimethyl suofoxidein and measured using a spectrophotometer (UV759CRT, Youke Instruments and Apparatus Inc., Shanghai, China) at 663 nm, 645 nm, and 470 nm using the following formulas [15,43]:(1)$$C_{a}=12.21\times A_{663}-2.81\times A_{645}$$
(2)$$C_{b}=20.13\times A_{645}-5.03\times A_{663}$$
(3)$$C_{caro}=\frac{1000\times A_{470}-3.27\times C_{a}-104\times C_{b}}{229}$$
where, *C_a_*, *C_b_*, and *C_caro_* are concentrations for chlorophyl-a, chlorophyl-b, and carotenoid and where *A*_663_, *A*_645_, and *A*_470_ are absorbances at 663 nm, 645 nm, and 470 nm, respectively. Protein content was determined using the Folin method at absorbance of 650 nm [20]. GS and ACP activities were assessed by methods adapted from Wei et al. [44]. Fine roots were kept in moist towels until scanning to obtain the projected image (HP Deskjet 1510 scanner, HP Inc., Palo Alto, CA, USA) at dots per inch of 118.11 pixels cm^−1^ [45]. Root images were analyzed using the WinRhizo software (Regent Instrument Inc., Canada) to assess fine root length, surface-area, average diameter, and tip number.

### 2.6. Variable Calculation and Statitstical Analysis

Root foraging precision was assessed by the absolute difference of fine root variables in two pot halves [27,28]. Precision assessment included variables of fine root difference of dry mass (FRMD), length (FRLD), and surface-area (FRSD) [24]. As the foraging scale also changes in response to light spectra, the absolute difference may be biased [27]. To cope with this bias, we also calculated the relative difference for dry mass (RFRMD), length (RFRLD), and surface-area (RFRSD) by dividing the absolute difference by the total amount of all both halves.

Shoot variables or traits (growth, biomass, nutrient concentrations, foliar physiology, and enzyme activity) were analyzed by a two-way analysis of variance (ANOVA). Therein, the type of light spectra was taken as the main-block source of variance and the pattern of nutrient availability (homogeneous vs. heterogeneous) was taken as the sub-block factor with three replicates of combined treatments. Root variables (morphology, biomass, and nutrient concentration) were analyzed by another two-way ANOVA, but the sub-block factor was replaced by nutrient availability of the pot-half patch. Either low or high availabilities were replicated three times in accordance with the heterogenous pattern, but the middle availability was replicated six times in accordance with both halves of the homogeneous pattern. As root foraging precision was assessed on the basis of different nutrient patterns, the two-way ANOVA used for FRMD, FRLD, FRSD, RFRMD, RFRLD, and RFRSD concerned two factors of light spectra and nutrient pattern. When significant effect was indicated, the results were compared by the main effects of light spectra, nutrient pattern, or half-pot availability (*α* = 0.05). When an interactive effect (light × pattern or light × availability) was indicated, the results were compared by a one-way ANOVA with combined factors as the unique source of variance. Duncan test was used for the comparison of difference due to the uneven number of replicates in different availabilities.

## 3. Results

### 3.1. Shoot Growth, Biomass, and Nutirent Concentration

Either light spectra or heterogenous pattern had a main effect on shoot height, but their effect on RCD was not statistically significant (Table 1). Compared with seedlings subjected to the green and blue spectrum, those in the red spectrum were taller by 14.5% (Figure 2A). The heterogeneous nutrient pattern resulted in higher seedling height by 13.6% compared with the homogeneous pattern (Figure 2B). Root-collar diameter ranged from 0.18 cm to 0.25 cm with a standard deviation (SD) of 0.02 (Figure 2C,D).

Shoot biomass was higher in the R spectrum by 33.7% compared with that in the GB spectrum (Figure 2E). The heterogenous pattern resulted in greater shoot biomass by 31.2% relative to the homogeneous pattern (Figure 2F). Neither factor had any significant effect on the root-to-shoot-biomass ratio, which ranged from 0.24 to 0.35 with SD of 0.04.

Light spectra and nutrient pattern had an interactive effect on shoot N concentration (Table 1). Seedlings in the GB spectrum had higher shoot N concentration by 38.2% and 30.5%, respectively, compared with those in the R spectrum (Figure 3A). Either light spectra or heterogeneous pattern had a main effect on the shoot P concentration (Table 1). The R spectrum resulted in a decrease in shoot P concentration by 21.5% compared to the GB spectrum (Figure 3B). The heterogeneous pattern resulted in a 19.3% higher shoot P concentration relative to the homogeneous pattern (Figure 3C).

### 3.2. Foliar Variables

Chlorophyl-a concentration was not statistically different in seedlings exposed to the two types of light spectra (Table 1). However, the R spectrum resulted in lower chlorophyl-b concentrations by 16.0% and higher carotenoid concentrations by 14.5%, compared with the GB spectrum (Table 2). Nutrient heterogeneous pattern always had a significant effect on chlorophyl and carotenoid concentrations (Table 1). Compared to seedlings subjected to the homogeneous pattern, those in the heterogeneous pattern had higher chlorophyl-a, chlorophyl-b, and carotenoid concentrations by 48.7%, 41.4%, and 13.6%, respectively (Table 2).

Foliar protein concentration was higher in the GB spectrum by 42.6% compared with that in the R spectrum (Table 2). The heterogeneous nutrient pattern resulted in a higher protein concentration by 24.2% compared with the homogeneous pattern.

The R spectrum resulted in a decrease in foliar GS activity by 17.3% compared with the GB spectrum (Table 2). However, light spectra did not affect foliar ACP activity. The heterogeneous pattern resulted in higher GS and ACP activities by 30.4% and 45.2%, respectively, compared with the homogeneous pattern.

### 3.3. Fine Root Morphology, Biomass, and Nutrient Concentration

Light spectra and half-pot nutrient availability had an interactive effect on fine root length (Table 3). Fine root length was higher by 135–196% in seedlings subjected to high fertility in the R spectrum compared with others (Figure 4).

Light spectra had a main effect on the fine root surface area, average diameter, tip number, and biomass (Table 3). Compared to seedlings subjected to the GB spectrum, those exposed to the R spectrum had higher surface areas, average diameter, tip number, and biomass by 64.2%, 6.1%, 63.3%, and 46.5%, respectively (Figure 5A,C,E,G). These four variables generally showed an increasing trend with the increase in half-pot fertility (Figure 5B,D,F,H). High fertility resulted in higher fine root surface area and biomass than low and medium fertilities (Figure 5B,H). Seedlings in low fertility pots had lower fine root diameter than those in middle and high fertilities (Figure 5D).

Light spectra and half-pot fertility had an interactive effect on root N concentration (Table 3). Root N concentration was highest in seedlings subjected to high fertility in the GB spectrum, followed by that subjected to middle fertility in the GB spectrum and high fertility in the R spectrum (Figure 6A). The GB spectrum resulted in higher root P concentrations by 16.9% compared to the R spectrum (Figure 6B). Seedlings subjected to high fertility had higher root P concentrations compared to those in low and middle fertilities (Figure 6C).

### 3.4. Root Foraging Precision

Variables of FRMD and RFRMD showed no significant response to light spectra, heterogeneous nutrient pattern, or their interaction (data not shown). However, light spectra and heterogeneous pattern had interactive effects on FRLD, FRSD, RFRLD, and RFRSD (Figure 7). All four variables were the highest in seedlings subjected to the heterogeneous nutrient pattern in the R spectrum.

## 4. Discussion

### 4.1. Shoot Response

We found that both shoot height and biomass were promoted by the red-light enriched spectrum compared to the blue-and-green light spectrum. This was not the first time that the promotion of the red-light enriched spectrum on shoot elongation and biomass accumulation was reported. Previous studies also revealed similar promotions on light-adapted species, such as *Bletilla striata* [10], *Pinus koraiensis* [16,20], and *Larix principis-rupprechtii* [15]. As was mentioned above, *P. pumila* is also a sunlight-adapted species and its shade tolerance is disputed [34,38,39]. In contrast, the spectrum enriched with red light did not clearly promote growth in deep-shade-tolerant species. For example, as a semi-shade-tolerant species, *Aralia elata* did not respond to the nature-based red-light enriched spectrum in either height growth or shoot biomass [8,21]. No matter the spectra used, stem diameter growth was also unaffected by spectra in *P. pumila*, *Dalbergia odorifera* [19], *Picea abies*, *Pinus sylvestris* [46,47], and *Quercus ithaburensis* var. *macrolepis* [22].

The heterogenous nutrient pattern resulted in an overall promotion of shoot development and nutrient uptake. Trials on three tropical tree species (*Cunninghamia lanceolata*, *Pinus massoniana*, and *Schima superba*) revealed contrasting results [48,49]. It was reported that, compared with the homogeneous pattern, trees in the heterogeneous pattern showed depressed above-ground growth and biomass both with P [49] or N [48] as the factor to drive changes. Some authors argued that the heterogenous pattern caused more photosynthetic products to be partitioned to underground organs, which lowered the growth of above-ground organs. However, another study on poplar stocks revealed that shoot growth was increased or unchanged in the heterogeneous pattern relative to the homogeneous pattern of Boron supply [50]. This was caused by the tradeoff between adjusting Boron deficiency and toxicity by signals within whole-plant cycling. As a dwarf species, *P. pumila* was able to maintain a slow growth rate in soils of alpine forests with strong heterogeneous patterns [41]. The homogeneous soil pattern may further restrict the development of shoots due to limited nutrient uptake that supports dry mass production.

### 4.2. Nutrient Uptake and Allocation

Interestingly, more N was absorbed and allocated to shoots in the heterogeneous nutrient pattern with the green-and-blue light spectrum compared with that with the red-light enriched spectrum. In red light, foliar chlorophyl content was lower or unchanged compared to that in green-and-blue light; therefore, photosynthetic pigment was not enhanced by the red-light enriched spectrum. In addition, foliar protein and N assimilation were also lowered in the red-light enriched spectrum, suggesting that N uptake demand was weakened. All of the above are reasons to explain why root N concentration was lowered in the red-light enriched spectrum in middle and high fertilities. All of these results concur with those found for *P. koraiensis* seedlings [20] but disagree with findings on *L. principis-rupprechtii* [15]. Regarding promoted shoot growth and biomass in the red-light enriched spectrum, it is reasonable to surmise that, compared with the green-and-blue light spectrum, the spectrum in red light induced an acceleration of N consumption for growth without any contribution to enhancing new uptake in *P. pumila*.

The foliar P content was not distinguished between the two types of light spectra although, again, P concentration was lower in the red-light enriched spectrum in both aboveground and belowground organs. This suggests that the red-light enriched spectrum restricted not only P uptake but also P allocation pattern. During this process, the red-light spectrum did not have any impact on assimilation as a driving force. Lower P uptake in the red-light enriched spectrum was also reported on different species of tree [20], shrub [21], and herb [10]. While some authors of studies on model plants have argued that red light may theoretically benefit P uptake through phytochrome B signaling mediation [51,52], it is unlikely that the mechanism can account for other natural plants due to a wide variation in expression levels of genes responsive to P starvation [53]. We surmise that the biomass increment resulted in a diluted P concentration in plants subjected to the red-light enriched spectrum without additional P supply.

Both N and P concentrations were higher in the roots of seedlings subjected to the heterogeneous nutrient pattern. Furthermore, root N and P concentrations increased with the increase in half-pot fertilities. This suggests higher foraging precision when absorbing nutrients in nutrient enriched patches than in nutrient poor patches. Our results concur with the findings on trees of *Ailanthus altissima* [54], *Solidago altissima*, *Pinus taeda*, *Liquidambar styraciflua* [55], *Pinus massoniana*, *Schima superba*, *Liriodendron chinese*, and *Cunninghamia lanceolata* [56]. As a result, the heterogeneous nutrient pattern resulted in higher shoot N and P concentrations compared to the homogeneous pattern due to stimulated nutrient uptake in enriched patches.

### 4.3. Root Foraging Precision

We can accept our first hypothesis because the red-light enriched spectrum promoted root foraging precision only in the heterogeneous nutrient pattern. The homogeneous pattern did not induce any response of foraging precision in both types of spectra. These results were derived from the effect fine root elongation with, accordingly, the increases in surface-area, diameter, and dry mass. The fine root surface area presented a similar response to combined spectra and heterogeneity factors to that for length. The surface area is the product of length and diameter, which both increased in high fertilities and red-light enriched spectrum. Precision was also found to be higher in the heterogeneous nutrient pattern [49], where root length was more responsive to partial nutrient availability in a heterogeneous soil environment than any other morphological parameters [24,28,29]. The lack of a response to fertility in the number of root tips suggests that fine root proliferation resulted from root elongation and diameter growth rather than the production of new lateral roots. In a study on two highly valued ornamental tree species, root tip number only increased with increasing partial nutrient supply for species with higher growing speed [24]. In another study testing root architecture in *Picea abies* families, tip number was also found to be significant in fast-growing families [57]. Therefore, we surmise that, due to the slow growing speed of *P. pumila*, the response of the root tip number was not significant. Further work is suggested to compare the tip numbers of *P. pumila* with other pine species with varied growing speeds to verify our assumption.

We can also accept our second hypothesis because the red-light enriched spectrum can promote fine root growth and morphology compared with that enriched with green and blue lights. However, the red-light enriched spectrum was found to depress fine root length in *Eleutherococcus senticosus* seedlings [14]. Studies on frontier tree species even reported no response of fine root length between different types of spectra [17,22]. Considering that the red-light enriched spectrum gives higher energy as more photons to plants relative to the green-and blue-light spectra in the same leaf area at a same time, our red-light enriched spectrum can be taken as a high-energy input. *Juniperus virginiana* has a similar ecological habit to that of *P. pumila* as both can dwell in shaded conditions despite a low shade tolerance. It was also found that unshaded plants receiving high energy also had larger fine root morphology in accordance with higher diameter [58].

Either light spectra or half-pot fertility had a main effect on root biomass in our study. According to previous reports, however, root dry mass has varied responses to light spectra. The red-light enriched spectrum was reported to increased root biomass in *Larix principis-rupprechtii* [15], *Pinus koraiensis* [16], *Pinus sylvestris*, *Abies borisii-regis* [17], and *Quercus ithaburensis* var. *macrolepis* [22]. It was also reported that the red-light enriched spectrum cannot change root biomass in *Acer truncatum*, *Quercus mongolica* [18], *P. koraiensis* [20], and *Dalbergia odorifera* [19]. As our green-blue light spectrum was designed by simulating sunlight parameters in undergrowth of natural forests, *P. pumila* acclimated to the irradiation that keeps root plasticity at a mild level. The red-light enriched spectrum was not derived through simulations of natural sunlight in habitats; hence, it promoted photosynthetic production for shoots and induced more dry mass partitioning to roots. The root foraging precision assessed by dry mass did not respond as it did in the fine root lengths and surfaces. This was caused by the null response to the interactive effects of light spectra and patchy fertility on root biomass. The fine root difference of dry mass was found to be different mostly among different co-existing herbal species [27,28]. It was also reported to be not as responsive to exogeneous stimuli as the fine root morphology in *Podocarpus macrophyllus* and *Taxus cuspidata* [24]. Overall, we recommend using fine root morphology rather than dry mass to assess root foraging precision for *P. pumila*.

## 5. Conclusions

In a study testing the response of *P. pumila* to simulated light spectra and heterogeneous substrate conditions in an all-controlled condition, we found that root foraging precision can be enhanced in seedlings subjected to the red-light enriched spectrum in a heterogenous nutrient pattern. This was due to the fine root morphology, biomass, and nutrient concentration promoted in nutrient-enriched patches. Compared with the green-and-blue light spectrum, the red-light enriched spectrum induced greater growth but lowered nutrient uptake and allocation. We conclude that *P. pumila* has a strong ability to forage nutrients in heterogenous soil nutrients by placing longer fine roots in nutrient-enriched patches. Light spectrum can generate an interactive effect on root foraging behavior with a higher red-light proportion.

## Figures and Tables

**Figure 1 plants-10-01482-f001:**
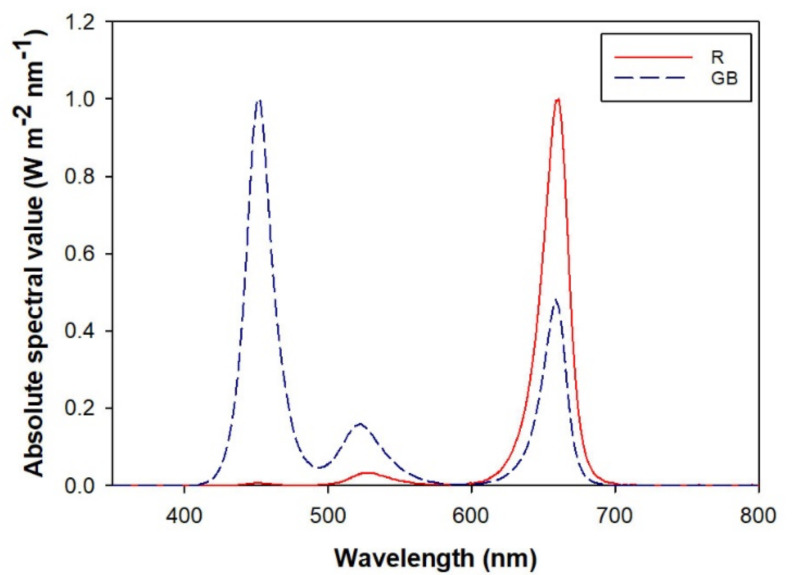
Spectral characteristic curves of red-light enriched (R) and green-and-blue light (GB) spectra.

**Figure 2 plants-10-01482-f002:**
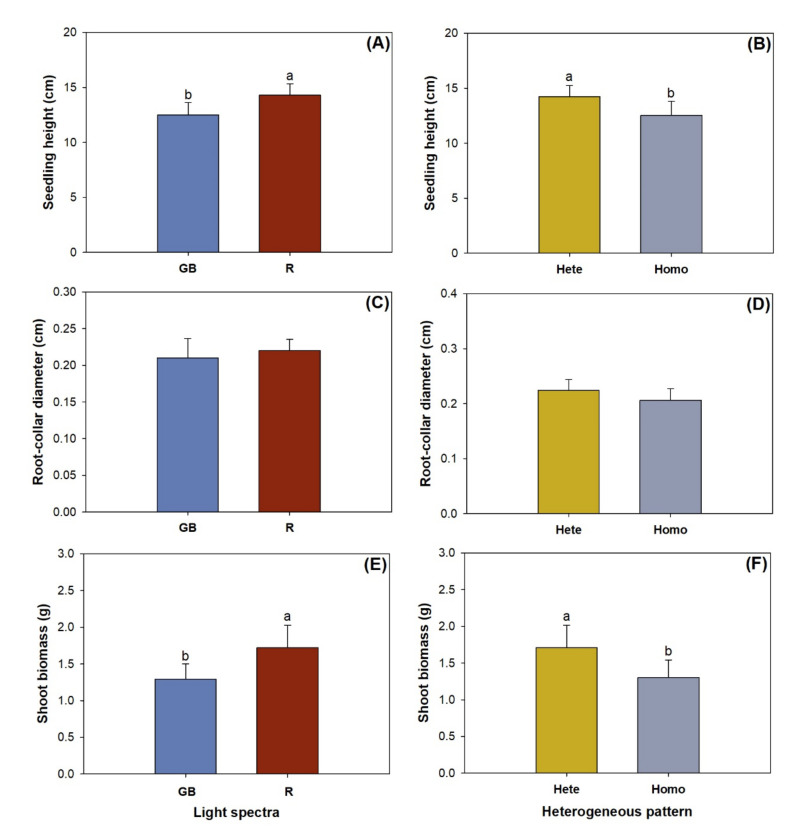
Seedling height (**A**,**B**), root-collar diameter (**C**,**D**), and shoot biomass (**E**,**F**) in *Pinus pumila* seedlings subjected to green-and-blue light (GB) and red-light enriched spectra (R) in heterogeneous (Hete) and homogeneous (Homo) nutrient patterns. Error bars indicate standard deviations. Different letters above the bars indicate significant differences according to Duncan test at the 0.05 level.

**Figure 3 plants-10-01482-f003:**
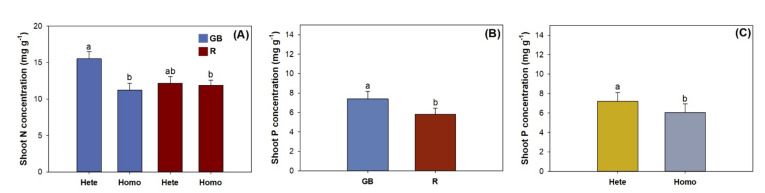
Nitrogen (N) and phosphorus (P) concentrations in *Pinus pumila* seedlings subjected to green-and-blue light (GB) and red-light enriched spectra (R) in heterogeneous (Hete) and homogeneous (Homo) nutrient patterns. Error bars indicate standard deviations. Different letters above bars indicate significant differences according to Duncan test at the 0.05 level. (**A**) shoot N concentration of seedlings subjected to combined light spectra and nutrient pattern; (**B**) shoot P concentration in response to light spectra; (**C**) shoot P concentration in response to combined nutrient pattern.

**Figure 4 plants-10-01482-f004:**
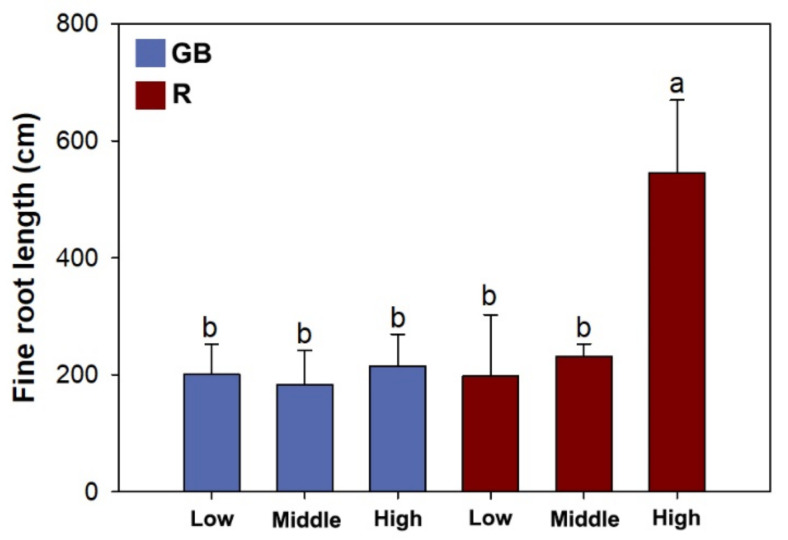
Fine root length in *Pinus pumila* seedlings subjected to green-and-blue light (GB) and red-light enriched spectra (R) in low, middle, and high half-pot fertilities. Error bars indicate standard deviations. Different letters above bars indicate significant difference according to Duncan test at the 0.05 level.

**Figure 5 plants-10-01482-f005:**
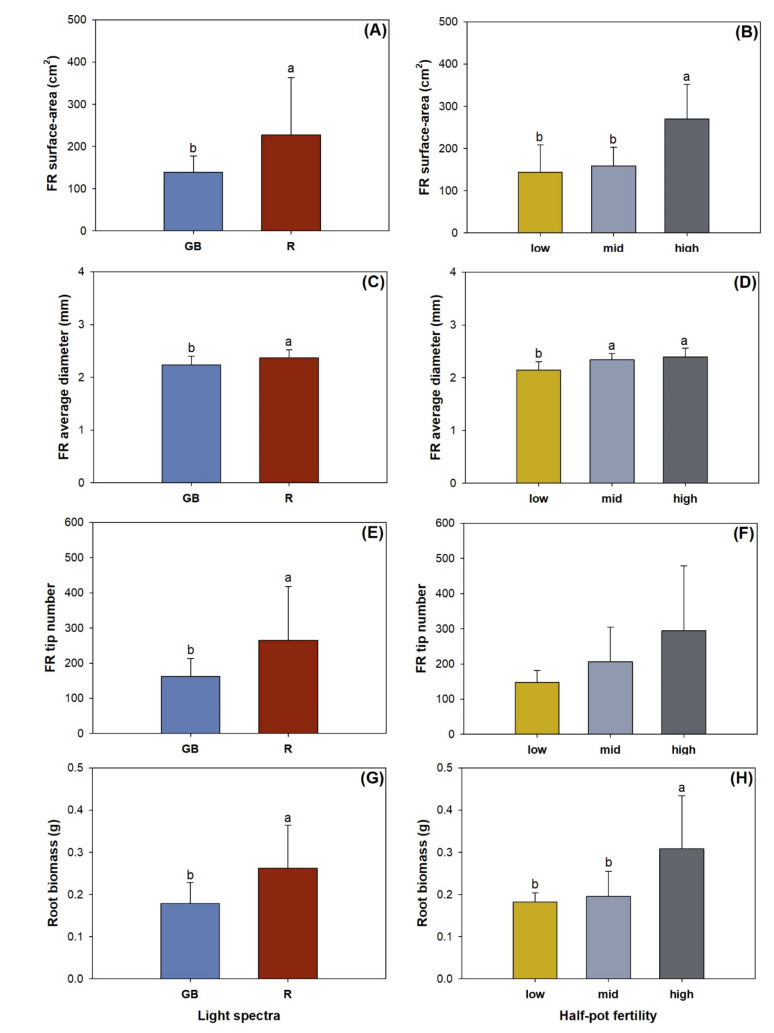
Fine root (FR) surface area (**A**,**B**), average diameter (**C**,**D**), tip number (**E**,**F**), and biomass (**G**,**H**) in *Pinus pumila* seedlings subjected to green-and-blue light (GB) and red-light enriched spectra (R) in low, middle, and high half-pot fertilities. Error bars indicate standard deviations. Different letters above the bars indicate significant differences according to Duncan test at the 0.05 level.

**Figure 6 plants-10-01482-f006:**
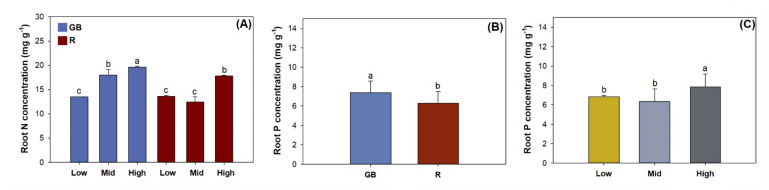
Nitrogen (N) and phosphorus (P) concentrations in *Pinus pumila* seedlings subjected to green-and-blue light (GB) and red-light enriched spectra (R) in low, middle, and high half-pot fertilities. Error bars indicate standard deviations. Different letters above the bars indicate significant difference according to Duncan test at the 0.05 level. (**A**) root N concentration of seedlings subjected to combined light spectra and half-pot fertility; (**B**) root P concentration in response to light spectra; (**C**) root P concentration in response to half-pot fetility.

**Figure 7 plants-10-01482-f007:**
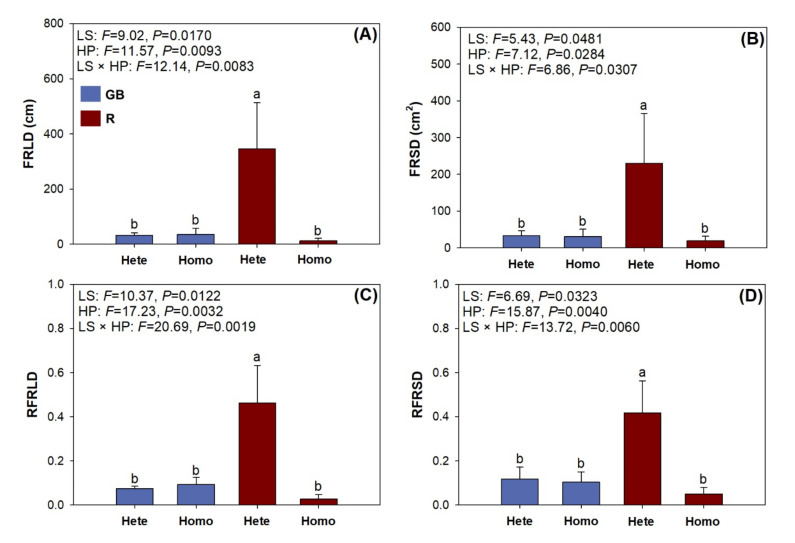
Fine root differences between pot-halves for length (FRLD) and surface area (FRSD), and their relative values (RFRLD and RFRSD, respectively) in *Pinus pumila* seedlings subjected to green-and-blue light (GB) and red-light enriched spectra (R) in heterogeneous (Hete) and homogeneous (Homo) nutrient patterns. Error bars indicate standard deviations. Different letters above the bars indicate significant difference according to Duncan test at the 0.05 level. (**A**) FRLD, (**B**) RFSD, (**C**) RFRLD, and (**D**) RFRSD in seedlings subjected to combined nutrient pattern and light spectra.

**Table 1 plants-10-01482-t001:** *F* values from the analysis of variance (ANOVA) of light spectra (LS), heterogeneous pattern (HP), and their interaction (LS × HP) on shoot growth, biomass, nitrogen (N) and phosphorus (P) concentrations and contents, and foliar variables in *Pinus pumila* (Pall.) Regel seedlings.

Variables	Source of Variance		
	LS	HP	LS × HP
Seedling height	24.21 * ^1^	21.43 *	0.07
RCD ^2^	0.78	2.26	0.88
Shoot biomass	24.42 *	21.49 *	0.15
R/S ^3^	0.59	0.04	0.07
Shoot N concentration	6.82 *	19.42 *	15.13 *
Shoot P concentration	84.28 ***	45.12 **	0.01
Foliar chlorophyl-a	0.56	19.33 *	2.00
Foliar chlorophyl-b	6.72 *	26.05 **	0.83
Foliar carotenoid	24.21 *	21.43 *	0.07
Foliar protein	16.75 *	6.34 *	3.61
GS activity ^4^	10.43 *	20.23 *	0.04
ACP activity ^5^	1.40	13.02 *	1.32

Note: ^1^ Asterisks indicate significant effect: * *p* < 0.05, ** *p* < 0.01, *** *p* < 0.0001; ^2^ RCD, root-collar diameter; ^3^ R/S, root-to-shoot-biomass ratio; ^4^ GS, glutamine synthetase; ^5^ ACP, acid phosphatase.

**Table 2 plants-10-01482-t002:** Foliar variables in *Pinus pumila* (Pall.) Regel seedlings subjected to contrasting light spectra and heterogeneous patterns.

Variable	Light Spectra		Heterogeneous Pattern	
	GB ^1^	R ^2^	Hete ^3^	Homo ^4^
Chlorophyl-a (mg g^−1^)	1.70 ± 0.55a ^5^	1.59 ± 0.27a	1.96 ± 0.33A	1.32 ± 0.16B
Chlorophyl-b (mg g^−1^)	1.81 ± 0.44a	1.52 ± 0.26b	1.95 ± 0.29A	1.38 ± 0.18B
Carotenoid (mg g^−1^)	12.48 ± 1.14b	14.29 ± 1.04a	14.24 ± 1.00A	12.53 ± 1.27B
Protein (mg g^−1^)	1.50 ± 0.32a	1.05 ± 0.16b	1.42 ± 0.40A	1.14 ± 0.22B
GS ^6^ activity (µgNPP g^−1^ FW min^−1^)	3.08 ± 0.48a	2.55 ± 0.48b	3.18 ± 0.37A	2.44 ± 0.41B
ACP ^7^ activity (A mg^−1^ protein h^−1^)	1.68 ± 0.54a	1.90 ± 0.39a	2.21 ± 0.28A	1.46 ± 0.37B

Note: ^1^ GB, combined green and blue light spectra; ^2^ R, red light spectrum; ^3^ Hete, heterogeneous pattern; ^4^ Homo, homogeneous pattern; ^5^ different letters indicate significant difference in contrasting conditions, lower case letters a and b label difference for light spectra and capital letters indicate differences for heterogeneous patterns; ^6^ GS, glutamine synthetase; ^7^ ACP, acid phosphatase.

**Table 3 plants-10-01482-t003:** *F* values from analysis of variance (ANOVA) of light spectra (LS), half-pot fertility (HPF), and their interaction (LS × HPF) on root morphology, biomass, and N and P concentrations and contents in *Pinus pumila* (Pall.) Regel seedlings.

Variables	Source of Variance		
	LS	HPF	LS × HPF
FR ^1^ length	7.88 * ^2^	8.11 *	6.04 *
FR surface-area	7.17 *	4.63 *	3.12
FR diameter	7.07 *	6.43 *	1.08
FR tip number	6.09 *	3.20	1.55
Biomass	10.59 *	7.97 *	1.07
N concentration	85.68 ***	59.69 ***	25.60 ***
P concentration	6.75 *	4.45 *	1.71

Note: ^1^ FR, fine root; ^2^ asterisks indicate significant effect: * *p* < 0.05, *** *p* < 0.0001; ^2^ RCD, root-collar diameter.

## Data Availability

Data is contained within the article.

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
