# Peer review of "Root Foraging Precision of Pinus pumila (Pall.) Regel Subjected to Contrasting Light Spectra"

_plants, 2021, doi:10.3390/plants10071482_

Round 1

Reviewer 1 Report

Dear Authors,

The manuscript “Root foraging precision of Pinus pumila (Pall.) Regel subjected to contrasting light spectra”, reviewed by me for "Plants", contains interesting and novel scientific data; generally, the experiments were well designed (with one reservation mentioned below) and the results are very interesting.
I regret that the experimental pine plants were not supplemented with mycorrhiza: since this symbiosis is practically obligatory for pines in nature, its exclusion makes the results less valuable as they cannot explain correctly root foraging under natural conditions. The results would also be of little use in improving the production of pine seedlings in forest nurseries, as mycorrhization is now a good practice in intensive container production. Hopefully, you will carry out appropriate experiments in the future.
Nevertheless, the manuscript should be published, after completing some of the minor data in the Materials and methods chapter and revising the English grammar and style. Try to write in a simpler style, because - apart from some grammatical mistakes - your text is difficult to read.
All my requirements, suggestions and comments are marked in the annotated manuscript file appended to the review.

With best regards, and best wishes,
Sincerely yours,

Reviewer

Author Response

The manuscript “Root foraging precision of Pinus pumila (Pall.) Regel subjected to contrasting light spectra”, reviewed by me for "Plants", contains interesting and novel scientific data; generally, the experiments were well designed (with one reservation mentioned below) and the results are very interesting.

RESPONSE: Thanks for the positive comments on our manuscript.

I regret that the experimental pine plants were not supplemented with mycorrhiza: since this symbiosis is practically obligatory for pines in nature, its exclusion makes the results less valuable as they cannot explain correctly root foraging under natural conditions. The results would also be of little use in improving the production of pine seedlings in forest nurseries, as mycorrhization is now a good practice in intensive container production. Hopefully, you will carry out appropriate experiments in the future.

RESPONSE: Thanks for suggesting about the mycorrhiza tissues. Yes, mycorrhiza is an important symbiosis with pine populations in natural habitats. It is a significant symbiont with pines and studies have also revealed its communities in Pinus pumila populations [1,2]. The ectomycorrhizal (ECM) fungi might be obligatory in the alpine pine ecosystem, but we do not see the obligation of ECM contribution to impose inevitable effects on root foraging behaviors. As we described in our study, foraging is mainly driven by the uneven distribution of nutrients in soils which generates a heterogeneous distribution pattern. There is no other things that can impose more powerful effect on fine root proliferation in fine sites rather than nutrients as soon as the aim of detecting foraging also matches the obligate objective driver. A study revealed that ECM fungal communities in P. pumila populations can be significantly affected by soil properties [2]. This clearly means that the diversity and structure of ECM community functioned as one of the responsers to soil properties, but this cannot be used as an instance that ECM can affect soil properties that can influence foraging inversely. Furthermore, ECM communities highly correspond to summer temperature [1] which also demonstrates that ECM is a responser to abiotic condition but not a driver. Finally, it was stated that ECM colonization of seedlings is critical to forest regeneration [2]. However, the stage of establishment was not the main objective of this study and we focused on the general response of root foraging behavior in P. pumila with seedlings as a model material. Foraging is not just found during the establishment stage (even we did not find any evidence about this issue). Instead, it is a common response to heterogeneous soil environment for mature tree tress. Hence, we admit that ECM is important for P. pumila populations but ECM is not more necessary than the direct test on soil nutrient supply in heterogeneous soils.

Authors also highly appreciate about the expection of ECM incubation for nursery cultured P. pumila seedlings. Thus, we used P. pumila seedlings as our materials in this study, but our objective was to simulate a natural response with juvenile plants just as materials. We did not concern anything to give application from our results to guide the practice in seedling culture of this species. This is because root foraging is not an eager issue for nursery manager, who will care more about the precise practices of exponential fertilization, container type, transplant manipulation, and of course, the use of ECM colonization before transplant. However, probably, we will study the techniques that can promote seedling quality at the end of nursery culture for P. pumila in the future.

Nevertheless, the manuscript should be published, after completing some of the minor data in the Materials and methods chapter and revising the English grammar and style. Try to write in a simpler style, because - apart from some grammatical mistakes - your text is difficult to read.

RESPONSE: Thanks for giving endorsement for the potential chance of publication of our study. We will complete the minor data and promote the English language by turning to professional agency.

All my requirements, suggestions and comments are marked in the annotated manuscript file appended to the review.

RESPOSNE: We will revise the manuscript according to comments in attached PDF file and make responses directly therein. Please see our attached PDF file.

Reviewer 2 Report

The article is interesting, but it contains a few points to explain:

  1. What were the 0.45 l pots made of?
  2. How many day-old seedlings were they?
  3. How were the seedlings watered?

Is it a very big problem in pot testing?

  1. In what mineral fertilizers was NPK applied?
  2. Were fertilizers applied to the pot surface?
  3. How many repetitions, draw?
  4. How were the roots divided into individual fractions?
  5. Please provide the formulas for calculating the chlorophyll content in the methodology?
  6. Why was chlorophyll a + b not listed?
  7. What was the nutrient uptake?
  8. In conclusion, please write that these were tests under controlled conditions.

Author Response

The article is interesting, but it contains a few points to explain:

  1. What were the 0.45 l pots made of?

RESPONSE: They were made by plastic.

  1. How many day-old seedlings were they?

RESPONSE: Germination persisted for 3-4 months depending on the order of germinating individuals. The process of seedling growth persisted for 3 months.

  1. How were the seedlings watered?

RESPONSE: Seedlings were watered by subirrigation in tanks. We highlighted this methodology in text.

  1. Is it a very big problem in pot testing?

RESPONSE: No. Not at all. Pot testing is just an approach that measured root foraging in heterogeneous environments in drought, which should be the big issue.

  1. In what mineral fertilizers was NPK applied?

RESPONSE: Please refer to 2.2 Controlled release fertilizers were used as a formulation of 14-13-13.

  1. Were fertilizers applied to the pot surface?

RESPONSE: No, fertilizers were broadcasted to the surface of substrates at the depth of 4cm. This depth matches that in the first sentence of 2.2. New information added.

  1. How many repetitions, draw?

RESPONSE: Please refer to 2.4. Three combined treatment replicates.

  1. How were the roots divided into individual fractions?

RESPONSE: Dried samples were excised firstly and ground for the determination of total N and P concentrations.

  1. Please provide the formulas for calculating the chlorophyll content in the methodology?

RESPONSE: Dried samples were excised firstly and ground for the determination of total N and P concentrations. Formulas have been supplied.

  1. Why was chlorophyll a + b not listed?

RESPONSE: Results about chlorophyll a+b were not significant. In addition, the physiological significance and meaning of this variable were not relevant to the study.

  1. What was the nutrient uptake?

RESPONSE: It has been fixed by another reviewer. It is nutrient concentration actually. Changes made.

  1. In conclusion, please write that these were tests under controlled conditions.

RESPONSE: Texted.

Round 2

Reviewer 2 Report

I am pleased with the changes made to the article.